# Hyaluronic Acid with Bone Substitutes Enhance Angiogenesis In Vivo

**DOI:** 10.3390/ma15113839

**Published:** 2022-05-27

**Authors:** Solomiya Kyyak, Sebastian Blatt, Nadine Wiesmann, Ralf Smeets, Peer W. Kaemmerer

**Affiliations:** 1Department of Oral and Maxillofacial Surgery, University Medical Center Mainz, Augustusplatz 2, 55131 Mainz, Germany; solomiya.kyyak@unimedizin-mainz.de (S.K.); sebastian.blatt@unimedizin-mainz.de (S.B.); nwiesmann@uni-mainz.de (N.W.); 2Department of Otorhinolaryngology, University Medical Center Mainz, Langenbeckstraße 1, 55131 Mainz, Germany; 3Division “Regenerative Orofacial Medicine”, University Medical Center Hamburg-Eppendorf, Martinistrasse 52, 20246 Hamburg, Germany; r.smeets@uke.de; 4Department of Oral and Maxillofacial Surgery, University Medical Center Hamburg-Eppendorf, Martinistrasse 52, 20246 Hamburg, Germany

**Keywords:** hyaluronic acid, xenogenic bone-substitute material, biofunctionalization, angiogenesis, osteogenesis, chick-embryo chorioallantoic membrane assay

## Abstract

Introduction: The effective induction of angiogenesis is directly related to the success of bone-substitute materials (BSM) for maxillofacial osseous regeneration. Therefore, the addition of pro-angiogenic properties to a commercially available bovine bone-substitute material in combination with hyaluronic acid (BSM+) was compared to the same bone-substitute material without hyaluronic acid (BSM) in an in-vivo model. Materials and Methods: BSM+ and BSM were incubated for six days on the chorioallantoic membrane (CAM) of fertilized chicken eggs. Microscopically, the number of vessels and branching points, the vessel area and vessel length were evaluated. Subsequently, the total vessel area and brightness integration were assessed after immunohistochemical staining (H&E, alphaSMA). Results: In the BSM+ group, a significantly higher number of vessels (*p* < 0.001), branching points (*p* = 0.001), total vessel area (*p* < 0.001) as well as vessel length (*p* = 0.001) were found in comparison to the BSM group without hyaluronic acid. Immunohistochemically, a significantly increased total vessel area (*p* < 0.001 for H&E, *p* = 0.037 for alphaSMA) and brightness integration (*p* = 0.047) for BSM+ in comparison to the native material were seen. Conclusions: The combination of a xenogenic bone-substitute material with hyaluronic acid significantly induced angiogenesis in vivo. This might lead to a faster integration and an improved healing in clinical situations.

## 1. Introduction

Bone regeneration in oral and maxillofacial reconstructive surgery is a standard procedure, but remains challenging in patients with severe co-morbidities. Generally, autologous bone with its well-known osteogenic and osteoinductive properties is considered as the clinical gold standard. However, it shows donor-site morbidity and relatively high resorption rates [1]. Bone-substitute material (BSM) of xenogenic, allogenic, or alloplastic origin is most frequently used as valid clinical alternatives in their specific indications [2]. Among others, graft bacterial contamination, immunologic response as well as the unfavourable biocompatibility and biochemical characteristics of BSM are considered major limitations of this approach [3]. All BSM are processed non-cellular and, consequently, exhibit mainly osteoconductive properties. To overcome the lack of osteoinductive and osteogenic properties, a variety of biofunctionalization possibilities such as the addition of growth factors, autologous platelet concentrates [4,5,6,7,8,9,10,11,12] as well as hyaluronic acid [8,13,14] have been studied.

Hyaluronan, a naturally occurring biopolymer, constitutes an extracellular, pericellular matrix, which can be found intracellularly in all living organisms that remains identical among different species [15]. It has been applied in a variety of medical indications, such as for glial cell culture, for cutaneous wound healing, for skin regeneration, as an injectable dermal filler as well as in skin-care products [15,16], as a supplementation of joint fluid by arthritis [17,18,19,20], in ophthalmic surgery, for improved regeneration of surgical wounds, as a drug delivery agent [21], and, finally, in bone regeneration [22].

It is composed of a basic unit of two sugars, N-acetylglucosamine, and glucuronic acid, which can be polymerized into large macromolecules [15,23]. It can form an acid (hyaluronic acid, HA) and a salt (hyaluronate) [24]. HA is a predominant component of the extracellular matrix (ECM) [15]. Commonly, HA is extracted from rooster combs or by recombinant production in *Streptococcus bacterium* [25]. It is biocompatible [26], bacteriostatic [27], non-immunogenic [28], hygroscopic, homeostatic, viscous and water soluble [29]. Enzymatic degradation of HA occurs, among others, under the effect of hyaluronidase that splits the macromolecule of HA into short and long dimeric chains polymers [15]. In contrast to HA breakdown products, which prompt gene expression in macrophages, endothelial cells and eosinophils, thus initiating inflammatory response [30], in some studies high-molecular HA is presented to support cell inactivity and tissue integrity [31]. Furthermore, its organizes improved collagen deposition [32,33] and prompts fibroblast proliferation [32,34], exhibits higher viscosity and higher biocompatibility, as well as showing anti-inflammatory effects and decreasing interleukin (IL)-1b, IL-6, tumor necrosis factor-a, and prostaglandin E2 production [35].

In a previous work, evidence was provided that a combination of xenogenic BSM with HA enhanced osteoblasts viability, migration, and proliferation in vitro [8]. Besides osteogenic activity [13], a number of studies revealed a strong pro-angiogenic activity of high and low molecular weight HA [36], which is limited to the fragments of 4 to 25 disaccharides in length [37,38,39,40]. However, Zhao et al. reported that high-molecular HA inhibits angiogenesis and low-molecular weight HA vice versa [35]. Nevertheless, in this indirect manner, through increased angiogenic activity, HA may additionally promote osteogenesis [41], probably via the accumulation of osteogenic growth factors [41].

The chicken chorioallantoic membrane (CAM) assay is a well-proven and standardly used in-vivo model to monitor angiogenesis [42,43,44,45,46]. Among other preferences, the model has a low cost, simplicity, easy accessibility, and reliability [47]. Recently, the CAM assay was shown to be suitable to study angiogenic properties of different bone-substitute materials [48]. Thus, the aim of the study was to compare the additional pro-angiogenic effect of the combination of HA with a commercially available bovine BSM versus the same BSM without HA in vivo via the CAM model.

## 2. Materials and Methods

### 2.1. Bone-Substitute Materials

Two bone-substitute materials of bovine origin were analysed: cerabone^®^ (BSM, botiss biomaterials GmbH, Zossen, Germany, granularity: 1–2 mm) and cerabonePlus^®^ (BSM+, botiss biomaterials GmbH, Zossen, Germany, granularity: 1–2 mm). CerabonePlus^®^ is commercially pre-treated with non-cross linked high molecular weight HA with a molecular mass of 1.9–2.1 MDa. Both materials were pre-hydrated before application to the CAM with sterile saline solution according to manufacturer’s protocol. Briefly, to obtain a pasty mass, one drop of saline solution was applicated to the 0.03 g of each of the bone-substitute materials, in 5 min the material was applied into the CAM model.

### 2.2. Chick Embryo Chorioallantoic Membrane Assay

As previously described, fertilized white Leghorn chicken eggs (LSL Rhein-Main, Dieburg, Germany) were used for the CAM assay [9]. Briefly, eggs were incubated horizontally at 37 °C and 60% humidity in an egg incubator (Brutmaschine-Janeschitz GmbH, Hammelburg, Germany) as described before [43]. On day 3 of the embryo development, 5–6 mL of albumin from the egg was aspirated with a sterile 10 mL syringe. A square window was cut into the shell. Afterwards the eggs were further incubated under the same conditions. On day 7, after positive evaluation of the integrity and viability of the embryo [48], the respective BSM (each 0.03 g) was applied inside of a silicon ring (Elastics, Dentaurum, Ispringen, Germany) onto the CAM (BSM vs. BSM+; *n* = 24 per group, totally *n* = 48).

Subsequently, after further incubation for 6 days (embryo development day 13) pictures of the CAM were taken via digital microscopy (30× magnification, KEYENCE, Neu-Isenburg, Germany). The calibration was conducted by means of an 800 μm grid. The region of interest (ROI) of the first two squares (800 μm × 800 μm per square) near the silicon ring in the left middle of it were defined and analysed (Figure 1). The analysis of the vessel number (counting total number of vessel branches) and number of branching points was conducted manually. Further analysis included total vessel area, total vessel length, mean vessel thickness and number of branching points. For this purpose, manual counting as well as a specific software was used (v3.0.0, KML Vision GmbH, IKOSA AI Platform, Graz, Austria), as described before [46]. For immunohistochemical analysis, eggs were processed as following. Finally, the embryos were euthanized by cutting the main vessels.

### 2.3. Immunohistochemical Analysis

After the ring was removed, the CAM area within a silicon ring was fixed on day 13 in formaldehyde for 24 h, which was followed by paraffin embedding as previously described [48]. Then, sections of five micrometer were cut tangentially to the CAM surface. Afterwards, for further histopathologic assessment of the vessels, a hematoxylin and eosin stain (H&E, Merck, Darmstadt, Germany), α-smooth muscle actin stain (alphaSMA, Sigma-Aldrich, St. Louis, MO, USA) and CD105 (Biorbyt, Cambridge, England) antibody stain were performed according to manufacturer’s instruction as described before [9] (Figure 1 and Figure 2). Finally, pictures of the slices (2× magnification) were taken with the light microscopy (KEYENCE, Neu-Isenburg, Germany) and analysed via the BZ-II Analyzer (KEYENCE, Neu-Isenburg, Germany) software for average area of vessel staining (average size of stained area, pixels) and total area (summarized amount of stained area, pixels) by H&E and alphaSMA slices. In case of H&E slices, an additionally brightness integration (sum of the whole brightness intensity of all ROIs, quadratcount method, pixels) was counted (Figure 2). The total area of the slices with corresponding values was equalized to 100% of one standardized area to compare the groups.

### 2.4. Statistical Analysis

All data were converted into mean values with standard error of the mean (SEM) (for parametric data) and median values (for non-parametric data). The results were calculated up to the second decimal place and were visualized using bar charts with error bars. Firstly, Shapiro–Wilk test (SWT) was applied to test for normal distribution. Then, for normally distributed data, a two-sided Student’s *t*-test for paired samples (*t*-test) was employed. In case of non-normal distributions, Mann–Whitney test (MWT) was applied. Additionally, a Kruskal–Wallis rank sum test (KWT) was applied to compare all groups. A *p*-value of ≤0.05 was considered to be statistically significant.

## 3. Results

### 3.1. Chorioallantoic Membrane Assay

On day 6 after material application and, accordingly, at day 13 of embryo development, data evaluation was conducted. Here, vessel number was significantly higher in the BSM+ group in comparison to the BSM group (BSM+: median 34.5 vessels from branch to branch per ROI vs. BSM: 7 per ROI; *p* < 0.001). In addition, the number of vessel branching points did statistically significantly differ between the two groups (BSM: median 4 vs. BSM+: 20 branching points per ROI, *p* < 0.001, Figure 1, Table 1).

The total vessel area was significantly larger in BSM+ when compared to BSM (BSM+: 7.19 × 10^5^ ± 0.5 × 10^5^ pixels vs. BSM: 4.07 × 10^5^ ± 0.57 × 10^5^, *p* < 0.001; *t*-test; *p* ≤ 0.001; KWT). When comparing the vessel total length of the two groups, the value was significantly higher in the BSM+ (914.17 ± 42.08 pixels) group in comparison to BSM (560.0 ± 70.58 pixels, *p* ≤ 0.001). The vessel mean thickness did not significantly differ between groups (BSM: 35.5 × 10^3^ ± 21.1 × 10^3^ vs. BSM+: 30.34 × 10^3^ ± 2.27 × 10^3^
*p* = 0.1). For the branching points calculated by the software as a plausibility check, the statistically significant differences between the groups were confirmed (BSM: 10.46 ± 1.56 vs. BSM+: 23.88 ± 1.38, *p* < 0.001; Table 2).

### 3.2. Immunohistochemical Analysis of Angiogenesis

Comparison of the average vessel area in H&E (BSM: 3388.07 ± 539.16 vs. BSM+: 3089.16 ± 974.6, *p* = 0.8) and alphaSMA staining (BSM: 3104.85 ± 1559.54 pixels vs. BSM+ 3175.57 ± 586.96, *p* = 0.13) did not demonstrate any significant differences between the groups.

BSM+ did show a significantly higher amount of total vessel area in H&E (BSM: 8.24 × 10^5^ ± 1.91 × 10^5^ vs. BSM+: 24.43 × 10^5^ ± 6.32 × 10^5^; *p* < 0.001) and alphaSMA staining (BSM: 16.27 × 10^5^ ± 5.56 × 10^5^ vs. BSM+: 42.0 × 10^5^ ± 6.91 × 10^5^, *p* = 0.037). Lastly, a distinctive difference in brightness integration between both groups was found (BSM: 2.57 × 10^4^ ± 0.37 × 10^4^ vs. BSM+:8.09 × 10^4^ ± 3.24 × 10^4^; *p* < 0.05; Table 3).

## 4. Discussion

In the presented in-vivo study, the pro-angiogenic properties of BSM with and without HA were compared. As the major result, the group with HA presented a microscopic and immunohistochemically distinctive advantageous effect on angiogenesis.

It is known that HA regulates the interaction of fibronectin with collagen, aggregates proteoglycans, and can self-associate to a considerable degree [37]. Among the biological functions of HA, control of tissue hydration and water transport, numerous receptor-mediated roles in cell detachment, mitosis, migration, inflammation, and even tumor development and metastasis are discussed [16,49,50,51]. In addition, HA is likely to behave as a signalling molecule by interacting with cell surface receptors, influencing cell proliferation, migration, differentiation, and gene expression [52]. The most predominant receptor of HA is a transmembrane receptor CD44: the higher the molecular weight of HA, the more binding sites for CD44 [53]. Interestingly, the activation of angiogenesis occurs due to weakening of cell–cell and cell–matrix interactions. With electron microscopy, significantly more small vessels in ischemic areas of infarcted myocardium after application of HA in comparison to the control group without HA were observed [37]. Moreover, it was shown that in such diseases as rheumatoid arthritis, osteoarthritis, and diabetic retinopathy, more vessels are observed adjacent to a hyaluronate-rich fluid [37]. In addition, Jansen et al. showed that HA is not cytotoxic and shows good biocompatibility [26], is highly non-antigenic and non-immunogenic because of its high structural homology across species, and lacking interaction with blood components [28]. Thus, *Staphylococcus aureus*, *Streptococcus pyogenes et pneumoniae* and *Clostridium perfringens*, produce hyaluronidase for an antigenic camouflage, subsequently staying unrecognized by phagocytes of the immune system [54]. Further, besides reducing the production and activity of proinflammatory mediators as well as matrix metalloproteinases, HA affects the function of immune cells [55]. Moreover, HA induces the adhesion of bacteria to biomaterials, reducing the risk of infection [56]. In our study, we included HA with molecular weight 1.9–2.1 MDa. Of note is that the variety of functions of the exogenous HA depend on its molecular weight [20]. High-molecular-weight HA (2 × 10^4^–4.5 × 10^5^ Da) enhances gene expression in macrophages, eosinophils, certain epithelial cells, and, finally, endothelial cells [30].

Interestingly, in the CAM model at day 7 using bone substitutes in combination with HA, a persistent compact layout of interlaced bone lamellae with obvious spaces between them, similar to those of the alveolar bone, was observed [48]. It was hypothesized by the authors that undifferentiated mesenchymal cells can be considered similar to those found in alveolar bone. Thus, the authors suggested that the combination of BSM with HA seems to be capable of differentiating mesenchymal cells into the osteoprogenitor and osteoblastic lineage [48]. This suggestion, however, has been proven by Zou et al. [41,57]. Therefore, HA may be seen as a biomolecule of choice in bioregenerative medicine, including osseous regeneration, since it enhances cell adhesion, migration, and proliferation [58,59]. In recent clinical studies, it has been proven that hyaluronic acid has a positive effect on bone formation [14,60]. In our recent study, evidence was provided that HA activates osteoblasts in vitro [6,8]. In addition to cell activation, in the present study, a positive effect of HA on stimulation on vessel formation was discussed and presented. This is confirmed by West et al., who found that HA degradation products are pro-angiogenic and that the effect is limited to fragments of between 4 and 25 disaccharides in length [37]. Another study also proved that hyaluronic acid stimulate neovascularization [36]. Additionally, it was also shown that degradation of HA in developing or remodelling environment potentially not only produces fragments that are pro-angiogenic, but also removes the antiangiogenic activity of the innate hyaluronate, which also befit our study model [37].

In a previous study, a positive effect of products of HA on angiogenesis was shown [37]. In the present study, the CAM assay was assessed for observing the respective angiogenic activity of the BSM in combination with HA. This assay is considered a unique physiologically relevant tissue microenvironment for quantifying angiogenesis stimulation [61]. Serving as a gas exchange surface, the CAM is an extraembryonic membrane that subsequently forms a dense capillary network [43,44]. Development of the blood vessels within the CAM starts from the centre and moves out to the periphery [42]. Angiogenesis is described as a process of a new vessel forming from the pre-existing vessels. Vasculogenesis, in turn, occurs from blood islands and angioblasts [62,63,64]. Thus, the relative number of blood vessels per branching point should be an indicator of angiogenesis, namely, of the number of new blood vessels derived from pre-existing blood vessels [61]. Interestingly, pre-existing vessels tend to be larger and less branched [61]. Thus, in the present study, a vessel count, as well as a count of branching points were included. BSM+ groups showed more vessels and on average a smaller vessel diameter, thus presenting increased angiogenesis in comparison to non-treated BSM groups.

It is suggested that the vessel formation within the CAM continued rapidly until day 10 and was completed by day 11–14 of embryo development [65]. Furthermore, by day 13, the CAM stops growing, which on the whole could be advantageous to observe the effect of materials on angiogenesis [66]. Hence, we chose day 13 of embryogenesis for our study to compare a vessel formation between groups without the disturbance of rapid vessel development of embryo itself. On day 6 after material application, the combination of BSM with HA showed significantly higher numbers of branching points, pointing out subsequently induced angiogenic processes. In immunohistochemical analysis, CAM treated with BSM+ showed smaller vessel diameter than the BSM group. This may indicate that the vessels in BSM+ appeared later. That is in accordance with the immunohistochemical analysis, where H&E staining showed a significantly higher numbers of brightness integration in BSM+ group, accordingly indicating “young” vessels. Additionally, the average vessel area in both H&E and alphaSMA staining were almost on the same level in both groups. Conversely, the total vessel area showed significant differences. It, thus, confirms that the amount of vessel in the BSM+ group was enhanced, being represented by numerous small-diameter vessels in contrast to the BSM group. To the best of our knowledge, there is only one study that observed the effect of BSM in combination with HA on vessel formation in a CAM model. Here, HA with BSM, having been applied on the day 9 on the CAM, showed increased CAM vascularization on day 2 after material application. However, at day 4, no significant changes were observed. Nevertheless, the number of specimens was only five per study group, which—also due to the high variability of the mode—does not allow an adequate statistical interpretation [48].

It is unclear why capillary vessels were observed to grow into or over the material of the BSM groups to a greater extent in comparison with the BSM+ group. It could be hypothesized that, in the BSM+ group, unfavourable enzymatic degradants of HA might be involved. However, those did not affect the general positive influence on angiogenesis, although Zhao et al. suggested that high-molecular HA inhibits angiogenesis [35]. For now, it is evident that HA removes the antiangiogenic activity of the native hyaluronate by degradation and generates fragments that are angiogenic [37]. In addition, HA is suggested to have a certain affect in tumor angiogenesis [52,67]. Thereby, the natural presence of native and degraded hyaluronate is believed to regulate a normal and pathological neovascularization. Accurate understanding and further studies of this regulatory process may enable us to have a better understanding of angiogenesis in peculiar conditions [37].

HA is proven to be highly water-soluble [15,68,69]. This signifies that HA should act for a short period of time because of a quick dissolving process, although in our study we revealed changes on day 6. It may be caused by the decreased solubility of HA in combination with BSM or even by the storage and release kinetics of the bovine bone substitute. Thus, in other study groups, it was confirmed that the solubility of the HA may vary in different environments, as far as depending on the length of the chains [29], cross linking [70], pH and chemical modification [71], where the HA concentration seems to be irrelevant [15].

In accordance with other studies in the field of angiogenesis, one of the limitations of the present study is that new vessel formation is hardly distinguishable from a rearrangement of existing vessels because of the applied material [72]. Furthermore, the number of vessels could be affected by a nonspecific inflammatory reactions on the material, evoking a secondary vasoproliferative response that it was possible to recognize exceptionally via detailed histological studies [73]. Even so, due to the relative immunodeficiency of the CAM model, this might need further in-vivo experiments using other species.

## 5. Conclusions

In summary, it could be concluded that the combination of BSM with HA significantly induces angiogenesis in vivo. Therefore, biofunctionalization of xenogenic BSM with HA is feasible and can widen the indication for osseous regeneration procedures. Future prospective studies are much in need to assess this effect in the clinical approach.

## Figures and Tables

**Figure 1 materials-15-03839-f001:**
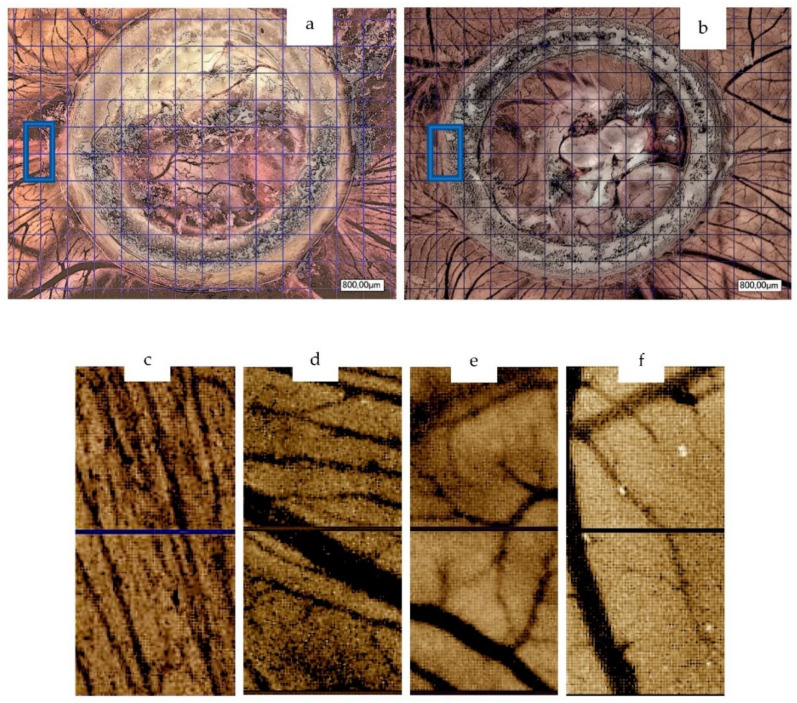
Microscope pictures (30× magnification, *n* = 24 per group) of BSM+ (**a**) and BSM (**b**) on day 6 of material application with indicated ROIs on CAM assay (30× magnification, 800 μm calibration, digital microscope KEYENCE, Neu-Isenburg, Germany); ROI in BSM+ (**c**,**d**) and BSM (**e**,**f**) by means of two calibrated squares.

**Figure 2 materials-15-03839-f002:**
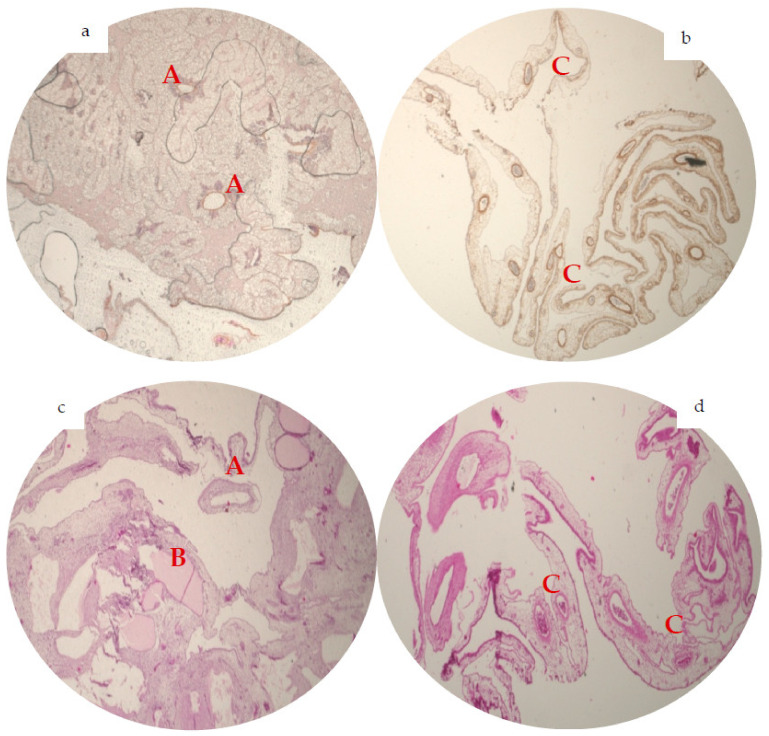
Representative immunohistochemical analysis of angiogenesis (2× magnification, *n* = 48). In alphaSMA staining, BSM group (**a**) showed solitary big-diameter vessels (A) and some small-diameter vessels. The alphaSMA staining of BSM+ (**b**) presented numerously small-diameter vessels, attaching prominently to the BSM+ (C). On the H&E staining, BSM (**c**) displays bigger diameter vessels (A) and almost no small-diameter vessels with a vessel grown through the BSM (B), in the H&E slices of BSM+ (**d**) a great number of a small- and medium-diameter vessels could be observed (C).

**Table 1 materials-15-03839-t001:** Number of vessels and branching point in BSM and BSM+ groups on day 6 by manual count (*n* = 24 for each group, two groups).

	BSM		BSM+	
	Vessels	Branching Points	Vessels	Branching Points
Median	7	4	34.5	20
MWT	0.00001	0.00001	0.00001	0.00001
KWT	0.00001	0.00001	0.00001	0.00001

**Table 2 materials-15-03839-t002:** Vessel total area, length, mean thickness and number of branching points counted on the chorioallantoic membrane assay on day 6 in BSM and BSM+ (*n* = 24 for each group, two groups).

	**BSM**			
	**Total Vessel Area**	**Total Vessel Length**	**Mean Vessel Thickness**	**Number of Branching Points**
Mean/median	4.07 × 10^5^ ± 0.57 × 10^5^ *	560 ± 70.58 **	35.5 × 10^3^ ± 21.1 × 10^3^ #	10.46 ± 1.56 ##
	**BSM+**			
	**Total Vessel Area**	**Vessel Total Length**	**Vessel Mean Thickness**	**Number of Branching Points**
Mean/median	7.19 × 10^5^ ± 0.5 × 10^5^ *	914.17 ± 42.08 **	30.34 × 10^3^ ± 2.27 ×10^3^ #	23.88 ± 1.38 ##
*t*-test	* *p* = 0.0001; ** *p* = 0.001			
KWT	# *p* = 0.091; ## *p* = 0.001			

**Table 3 materials-15-03839-t003:** Analysed H&E and alphaSMA staining for average area of vessel staining, total area, and brightness integration (only H&E) by the BSM and BSM+ groups on day 6 (*n* = 24 for each group, two groups).

**H&E**	**BSM**			**BSM+**		
	**Average Vessel Area**	**Total Vessel Area**	**Brightness Integration**	**Average Vessel Area**	**Total Vessel Area**	**Brightness Integration**
Mean/median	3388.07 ± 539.16 *	8.24 × 10^5^ ± 1.91 ×10^5^ **	2.34 × 10^4^ #	3089.16 ± 974.60 *	24.43 × 10^5^ ± 6.32 × 10^5^ **	3.12 × 10^4^ #
*T*-test	* *p* = 0.80; ** *p* = 0.0001					
MWT	# *p* = 0.047					
KWT	* *p* = 0.16; ** *p* = 0.001; # *p* = 0.05					
**alphaSMA**	**BSM**			**BSM+**		
	**Average Vessel Area**	**Total Vessel Area**		**Average Vessel Area**	**Total Vessel Area**	
Mean/median	1400.68 *	16.27 × 10^5^ ± 5.56 × 10^5^ **		3038.46 *	42.0 × 10^5^ **	
MWT	* *p* = 0.13; ** *p* = 0.037					
KWT	* *p* = 0.12; ** *p* = 0.036					

## Data Availability

Data are available on request.

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
