# Peer review of "Hyaluronic Acid with Bone Substitutes Enhance Angiogenesis In Vivo"

_materials, 2022, doi:10.3390/ma15113839_

Round 1

Reviewer 1 Report

Dear Authors,

Although not a novelty in the field, your article confirms that HA added to BSM induces angiogenesis on CAM model, which might improve healing in clinical situations, but without evaluating the inflammatory reactions induced by the material.

The article is well written, with appropriate Introduction and Materials and Methods, pertinent Results and Discussion section.

Minor improvement of English language and formatting are necessary for publication.

Further studies are really necessary to analyse how this phenomenon is clinically improving the healing of patients.

Author Response

  1. Reviewer comment: Although not a novelty in the field, your article confirms that HA added to BSM induces angiogenesis on CAM model, which might improve healing in clinical situations, but without evaluating the inflammatory reactions induced by the material.

Answer: Thank you for Your kind comment.

Although it was not proven in the present study, HA has been previously shown as not cytotoxic, highly non-antigenic and non-immunogenic. Consequently, as far as the BSM used in two groups was the same, it may be concluded that the difference in angiogenesis is not caused by the immune reaction. (In article: “..Besides, Jansen et al. showed that HA is not cytotoxic and shows good biocompatibility [26], is highly non-antigenic and non-immunogenic, because of its high structural homology across species, and lacking interaction with blood components [28]..”, line 235).

Moreover,  “..Thus, Staphylococcus aureus, Streptococcus pyogenes et pneumoniae and Clostridium perfringens, produce hyaluronidase for an antigenic camouflage, subsequently staying unrecognized by phagocytes of the immune system [51]. Further, besides reducing the production and activity of pro-inflammatory mediators as well as matrix metalloproteinases, HA affects the function of immune cells [52].., line 238”

However, because we didn’t prove an immune reaction in our article, that was described as one of the limitations, but rather as the limitation of the CAM model than the study itself. Because the CAM model is nevertheless also immunodeficient and is a bad example for studying immune reaction (Line 333).

Reviewer 2 Report

Dear authors , thank you for this interesting study. It certainly adds some new information about the effects of HA on blood vessel formation. 

However , some changes to the manuscript should be done. Please find my comments below. 

Your sentence line 66: In a previous work, evidence was provided that a combination of xenogenic BSM  with HA enhanced osteoblasts viability, migration, and proliferation in vitro [6]. I believe this should be reference 8? Reference 6 deals with PRF.  Please check it again.

Raines AL, Sunwoo M, Gertzman AA, Thacker K, Guldberg RE, Schwartz Z, Boyan BD. Hyaluronic acid stimulates neovascularization during the regeneration of bone marrow after ablation. J Biomed Mater Res A. 2011 Mar 1;96(3):575-83. doi: 10.1002/jbm.a.33012. Epub 2011 Jan 10. PMID: 21254389; PMCID: PMC4296566.

Please add this reference since it directly correlates to your topic of new vessel formation when HA is combined with a bone graft, and discuss their results with yours. This reference is for both the introduction and discussion section.

Božić D, Ćatović I, Badovinac A, Musić L, Par M, Sculean A. Treatment of Intrabony Defects with a Combination of Hyaluronic Acid and Deproteinized Porcine Bone Mineral. Materials (Basel). 2021 Nov 11;14(22):6795. doi: 10.3390/ma14226795. PMID: 34832196; PMCID: PMC8624958.

Pleases add this recent reference in the introduction since it shows that when HA is combined with a xenogeneic bone grafting material a greater amount of clinical attachment level can be observed, than when this particular HA is used alone if compared to the Pilloni study (Pilloni A, Rojas MA, Marini L, Russo P, Shirakata Y, Sculean A, Iacono R. Healing of intrabony defects following regenerative surgery by means of single-flap approach in conjunction with either hyaluronic acid or an enamel matrix derivative: a 24-month randomized controlled clinical trial. Clin Oral Investig. 2021 Aug;25(8):5095-5107. doi: 10.1007/s00784-021-03822-x. Epub 2021 Feb 10. PMID: 33565017; PMCID: PMC8342388.)

One of the reasons could be related to your findings of greater angiogenesis. Add this either in the introduction section or discuss these difference in the discussion part of the manuscript.   

Although in the introduction part you do mention that both high and low MW HA is angiogenic, please add more information about the differences between the both. High-MW HA exhibits higher viscosity, longer residence time, and higher biocompatibility. High-MW HA also shows anti-inflammatory effects, and inhibits angiogenesis, decreases interleukin (IL)-1b, IL-6, tumor necrosis factor-a, and prostaglandin E2 production. The Low-MW HA on the other hand has been reported to stimulate angiogenesis.

 In its natural state it is a HMW molecule.

The importance of different MWs is shown in this study: Zhao N, Wang X, Qin L, Guo Z, Li D. Effect of molecular weight and concentration of hyaluronan on cell proliferation and osteogenic differentiation in vitro. Biochem Biophys Res Commun. 2015 Sep 25;465(3):569-74. doi: 10.1016/j.bbrc.2015.08.061. Epub 2015 Aug 15. PMID: 26284973.

In the discussion part you should add some information about CD44, since it is the predominant receptor of HA. The higher the MW of HA, the more binding sites for CD44.  CD44 is a transmembrane receptor expressed by many cells, such as mesenchymal cells, and binding ligands, such as hyaluronic acid. Please address this issue in your model of angiogenesis.

Your sentence lines 240 and 241: Thus authors suggested that the combination of BSM with HA seem to be capable to differentiate mesenchymal cells into the osteoprogenitor and osteoblastic lineage [46].

There are other articles, one that you actually cite by Sasaki, that show that HA indeed induces differentiation of mesenchymal cells into the osteoblastic lineage and the one in the brackets (Zou L, Zou X, Chen L, Li H, Mygind T, Kassem M, Bünger C. Effect of hyaluronan on osteogenic differentiation of porcine bone marrow stromal cells in vitro. J Orthop Res. 2008 May;26(5):713-20. doi: 10.1002/jor.20539. PMID: 18050326.)

Please add these references to your sentence and rephrase the conclusion since it is not suggested but a proven fact.

Your sentence line 261 : Thus, the relative number of blood vessels per branching point should be indicator of the angiogenesis, namely of the number of new blood vessels derived from preexisting blood vessels [56].  Please add the word AN before the word indicator, actually is should be: should be an indicator of angiogenesis…….

Your sentence line 292: It could be hypothesized that in the BSM+ group unfavourable enzymatic degradants of HA might be involved. Though, those did not affect the general positive influence on angiogenesis.

This could be due to the MW as suggested before in the comments. You could address the MW here and the differences in their activities.

Author Response

  1. Reviewer comment: Your sentence line 66: In a previous work, evidence was provided that a combination of xenogenic BSM  with HA enhanced osteoblasts viability, migration, and proliferation in vitro [6]. I believe this should be reference 8? Reference 6 deals with PRF.  Please check it again.

Answer: Thank you for Your kind comment.

Correction is made according to the comments

  1. Reviewer comment: Raines AL, Sunwoo M, Gertzman AA, Thacker K, Guldberg RE, Schwartz Z, Boyan BD. Hyaluronic acid stimulates neovascularization during the regeneration of bone marrow after ablation. J Biomed Mater Res A. 2011 Mar 1;96(3):575-83. doi: 10.1002/jbm.a.33012. Epub 2011 Jan 10. PMID: 21254389; PMCID: PMC4296566.

Please add this reference since it directly correlates to your topic of new vessel formation when HA is combined with a bone graft, and discuss their results with yours. This reference is for both the introduction and discussion section.

Answer: Thank you for Your kind comment.

Citation is added in the Section Introduction line 73 and in the section Discussion Line 263

  1. Reviewer comment: Božić D, Ćatović I, Badovinac A, Musić L, Par M, Sculean A. Treatment of Intrabony Defects with a Combination of Hyaluronic Acid and Deproteinized Porcine Bone Mineral. Materials (Basel). 2021 Nov 11;14(22):6795. doi: 10.3390/ma14226795. PMID: 34832196; PMCID: PMC8624958.

Pleases add this recent reference in the introduction since it shows that when HA is combined with a xenogeneic bone grafting material a greater amount of clinical attachment level can be observed, than when this particular HA is used alone if compared to the Pilloni study (Pilloni A, Rojas MA, Marini L, Russo P, Shirakata Y, Sculean A, Iacono R. Healing of intrabony defects following regenerative surgery by means of single-flap approach in conjunction with either hyaluronic acid or an enamel matrix derivative: a 24-month randomized controlled clinical trial. Clin Oral Investig. 2021 Aug;25(8):5095-5107. doi: 10.1007/s00784-021-03822-x. Epub 2021 Feb 10. PMID: 33565017; PMCID: PMC8342388.)

Answer: Thank you for Your kind comment.

Thank you for an interesting study. It is added line 45 and line 258.

  1. Reviewer comment: Although in the introduction part you do mention that both high and low MW HA is angiogenic, please add more information about the differences between the both. High-MW HA exhibits higher viscosity, longer residence time, and higher biocompatibility. High-MW HA also shows anti-inflammatory effects, and inhibits angiogenesis, decreases interleukin (IL)-1b, IL-6, tumor necrosis factor-a, and prostaglandin E2 production. The Low-MW HA on the other hand has been reported to stimulate angiogenesis. The importance of different MWs is shown in this study: Zhao N, Wang X, Qin L, Guo Z, Li D. Effect of molecular weight and concentration of hyaluronan on cell proliferation and osteogenic differentiation in vitro. Biochem Biophys Res Commun. 2015 Sep 25;465(3):569-74. doi: 10.1016/j.bbrc.2015.08.061. Epub 2015 Aug 15. PMID: 26284973.

Answer: Thank you for Your kind comment.

Citation and discussion is added in line 69 and 74

  1. Reviewer comment: In the discussion part you should add some information about CD44, since it is the predominant receptor of HA. The higher the MW of HA, the more binding sites for CD44.  CD44 is a transmembrane receptor expressed by many cells, such as mesenchymal cells, and binding ligands, such as hyaluronic acid. Please address this issue in your model of angiogenesis.

Answer: Thank you for Your kind comment. The information is added in line 222.

  1. Reviewer comment: Your sentence lines 240 and 241: Thus authors suggested that the combination of BSM with HA seem to be capable to differentiate mesenchymal cells into the osteoprogenitor and osteoblastic lineage [46]. There are other articles, one that you actually cite by Sasaki, that show that HA indeed induces differentiation of mesenchymal cells into the osteoblastic lineage and the one in the brackets (Zou L, Zou X, Chen L, Li H, Mygind T, Kassem M, Bünger C. Effect of hyaluronan on osteogenic differentiation of porcine bone marrow stromal cells in vitro. J Orthop Res. 2008 May;26(5):713-20. doi: 10.1002/jor.20539. PMID: 18050326.) Please add these references to your sentence and rephrase the conclusion since it is not suggested but a proven fact.

Answer: Thank you for Your comment. The information is added in line 254.

  1. Reviewer comment: Your sentence line 261: Thus, the relative number of blood vessels per branching point should be an indicator of the angiogenesis, namely of the number of new blood vessels derived from preexisting blood vessels [56].  Please add the word AN before the word indicator, actually is should be: should be an indicator of angiogenesis…….

Answer: Thank you for Your kind comment. The corrections are added, actual line 276.

  1. Reviewer comment: Your sentence line 292: It could be hypothesized that in the BSM+ group unfavorable enzymatic degradants of HA might be involved. Though, those did not affect the general positive influence on angiogenesis. This could be due to the MW as suggested before in the comments. You could address the MW here and the differences in their activities.

Answer: Thank you for Your kind comment. The corrections are added, line 309.

Reviewer 3 Report

Title: Hyaluronic Acid for Biofunctionalization of Bone Substitutes to Enhance Angiogenesis in vivo

This study aimed to compare the additional pro-angiogenic effect of the combination of HA with a commercially available bovine BSM versus the same BSM without HA in vivo via the CAM model.

The study is very interesting, well-developed, and well-written.
After reading the text, some concerns were raised.

- I suggest modifying the title
- Explain better how were mixed the grafts and CAM
- Could the authors justify better why the results were better for the BSM+?
- Why did the authors affirm in the abstract conclusion (conclusion of the study) “xenogenic bone substitute material with hyaluronic acid significantly induced…” used the term hyaluronic acid? When was it obtained?

Author Response

  1. Reviewer comment: I suggest modifying the title

Answer: Thank You for Your suggestion. The title is modified

  1. Reviewer comment: Explain better how were mixed the grafts and CAM
    Answer: Thank You for Your comment. The corrections are in the line 93
  2. Reviewer comment: Could the authors justify better why the results were better for the BSM+

Answer: Thank You for Your comment.

In the Line 222 we discuss properties of HA

It is known that HA regulates the interaction of fibronectin with collagen, aggregates proteoglycans, and can self-associate to a considerable degree [38]. Among the biological functions of HA, control of tissue hydration and water transport, numerous receptor-mediated roles in cell detachment, mitosis, migration, inflammation, and even tumor development and metastasis are discussed [17, 50-52]. Also, HA is likely to behave as a signalling molecule by interacting with cell surface receptors, influencing cell proliferation, migration, differentiation, and gene expression [53]. The most predominant receptor of HA is a transmembrane receptor CD44: the higher the molecular weight of HA, the more binding sites for CD44 [54]. Interestingly, the activation of angiogenesis occurs due to weakening of cell-cell and cell-matrix interactions. With electron microscopy,  significantly more small vessels in ischemic areas of infarcted myocardium after application of HA in comparison to the control group without HA were observed [38]. Moreover, it was shown that in such diseases as rheumatoid arthritis, osteoarthritis, and diabetic retinopathy, more vessels are observed adjacent to a hyaluronate-rich fluid [38]. Besides, Jansen et al. showed that HA is not cytotoxic and shows good biocompatibility [27], is highly non-antigenic and non-immunogenic, because of its high structural homology across species, and lacking interaction with blood components [29].  Thus, Staphylococcus aureus, Streptococcus pyogenes et pneumoniae and Clostridium perfringens, produce hyaluronidase for an antigenic camouflage, subsequently staying unrecognized by phagocytes of the immune system [55]. Further, besides reducing the production und activity of proinflammatory mediators as well as matrix metalloproteinases, HA affects the function of immune cells [56]. Moreover, HA induces adhesion of bacteria to biomaterials, reducing the risk of infection [57]. In our study, we included HA with molecular weight 1.9–2.1 MDa. Of note, the variety of functions of the exogenous HA depend on its molecular weight [21]. High molecular weight HA (2 × 104–4.5 × 105 Da) enhances gene expression in macrophages, eosinophils, certain epithelial cells, and finally endothelial cells [31]. 

In the section Discussion Line 288 we discuss how we compared the groups

On day 6 after material application, the combination of BSM with HA showed significantly higher numbers of branching points, pointing out subsequently on induced angiogenic processes. In immunohistochemical analysis, CAM treated with BSM+ showed smaller vessel diameter than the BSM group. This may indicate that the vessels in BSM+ appeared later. That is in accordance with the immunohistochemical analysis, where H&E staining showed a significant higher numbers of brightness integration in BSM+ group, accordingly indicating “young” vessels. Additionally, the average vessel area in both H&E and alphaSMA staining were almost on the same level in both groups. Whereas the total vessel area showed significant differences. It, thus, confirms that the amount of vessel in the BSM+ group was enhanced, being represented by a numerous small-diameter vessel in contrast to BSM group.

Concerning the aforementioned I am not quite sure what the Reviewer means by “..justifying better why the results were better for the BSM+..”

  1. Reviewer comment: Why did the authors affirm in the abstract conclusion (conclusion of the study) “xenogenic bone substitute material with hyaluronic acid significantly induced…” used the term hyaluronic acid? When was it obtained?

Answer: I am afraid I do not fully understand the question. The HA was not obtained in the study, we used it as a commercially produced product together with the BSM. Does the Reviewer mean we should mention it in the conclusion? If yes, I would rather not do that, because the conclusion will be overfilled with the information and its point could be lost.

Round 2

Reviewer 3 Report

Title: Hyaluronic Acid for Biofunctionalization of Bone Substitutes to Enhance Angiogenesis in vivo

This study aimed to compare the additional pro-angiogenic effect of the combination of HA with a commercially available bovine BSM versus the same BSM without HA in vivo via the CAM model.

The study is very interesting, well-developed, and well-written.

Thank you for your revision.